# Ce-Doped Three-Dimensional Ni/Fe LDH Composite as a Sulfur Host for Lithium–Sulfur Batteries

**DOI:** 10.3390/nano13152244

**Published:** 2023-08-03

**Authors:** Huiying Wei, Qicheng Li, Bo Jin, Hui Liu

**Affiliations:** Key Laboratory of Automobile Materials, Ministry of Education, and School of Materials Science and Engineering, Jilin University, Changchun 130022, China; weihy22@mails.jlu.edu.cn (H.W.); liqc20@mails.jlu.edu.cn (Q.L.); liuhui20@mails.jlu.edu.cn (H.L.)

**Keywords:** lithium–sulfur batteries, layered double hydroxide, hollow capsule, shuttling effect, lithium polysulfides

## Abstract

Lithium–sulfur batteries (LSBs) have become the most promising choice in the new generation of energy storage/conversion equipment due to their high theoretical capacity of 1675 mAh g^−1^ and theoretical energy density of 2600 Wh kg^−1^. Nevertheless, the continuous shuttling of lithium polysulfides (LiPSs) restricts the commercial application of LSBs. The appearance of layered double hydroxides (LDH) plays a certain role in the anchoring of LiPSs, but its unsatisfactory electronic conductivity and poor active sites hinder its realization as a sulfur host for high-performance LSBs. In this paper, metal organic framework-derived and Ce ion-doped LDH (Ce-Ni/Fe LDH) with a hollow capsule configuration is designed rationally. The hollow structure of Ce-Ni/Fe LDH contains a sufficient amount of sulfur. Fe, Ni, and Ce metal ions effectively trap LiPSs; speed up the conversion of LiPSs; and firmly anchor LiPSs, thus effectively inhibiting the shuttle of LiPSs. The electrochemical testing results demonstrate that a lithium–sulfur battery with capsule-type S@Ce-Ni/Fe LDH delivers the initial discharge capacities of 1207 mAh g^−1^ at 0.1 C and 1056 mAh g^−1^ at 0.2 C, respectively. Even at 1 C, a lithium–sulfur battery with S@Ce-Ni/Fe LDH can also cycle 1000 times. This work provides new ideas to enhance the electrochemical properties of LSBs by constructing a hollow capsule configuration.

## 1. Introduction

With the sustainable development of the economy and society, people’s sense of crisis about fossil fuel exhaustion is increasing day by day. It has become an inevitable trend of future development to abandon our dependence on resources and develop advanced electrochemical energy storage/conversion systems. Lithium–sulfur batteries (LSBs) have been extensively studied by researchers due to their high theoretical capacity and theoretical energy density (Appendix A) [1,2]. Although LSBs possess broad prospects, they often encounter many troubles in practical applications, such as the poor electronic conductivity of active sulfur, the shuttling effect of lithium polysulfides (LiPSs), and the volumetric expansion of cathode materials during the charging/discharging processes.

To solve these problems, people have invested a lot of energy into the research and development of cathode materials in the past decades. So far, there are massive sulfur host materials, such as carbonaceous materials [3,4], metal compounds [5,6,7,8,9,10,11,12], and so on. Metal organic frameworks (MOFs) materials are excellent templates to synthesize hollow polyhedrons and hollow carbon materials by means of calcining or etching [13,14]. In recent years, layered double hydroxide (LDH) has been proven to adsorb LiPSs effectively in LSBs [15,16,17] and is also used in supercapacitors [18,19]. Due to sufficient hydrophilic points and hydroxyl groups [15], LiPSs are not easily dissolved in organic electrolytes, thus promoting the redox kinetics of LiPSs. A novel lithium–sulfur battery configuration was designed by Hwang et al. [15], which consists of a sulfur and magnesium–aluminum-layered double hydroxide (MgAl LDH) carbon nanotube composite material as the positive electrode and a modified separator. This configuration achieves high specific capacity and good capacity retention, and this positive electrode can be used for both the catalytic conversion of LiPSs and anchoring of LiPSs, thereby improving the rapid reaction kinetics and Coulombic efficiency (CE). Chen et al. [16] used ZIF-67 as a template to prepare a regular nanocapsid-shaped NiCo-LDH/Co_9_S_8_ (H-LDH/Co_9_S_8_) hollow structure to inhibit LiPS diffusion, and its hollow structure accommodates sufficient sulfur. Rich O and Co sites block the migration and diffusion of LiPSs through chemical bonding and promote the transformation kinetics of LiPSs. At 0.1 C, H-LDH/Co_9_S_8_ has a high initial capacity of 1339.1 mAh g^−1^. H-LDH/Co_9_S_8_ possesses an ultra-long lifespan of up to 1500 cycles (0.047% of the capacity decay rate per cycle, 1 C) and a stable CE of over 98%. This work opens up a new way for LDH materials to be used as sulfur carrier materials for high-performance LSBs. Zhang et al. [17] designed and synthesized nanocages with two shells of Co(OH)_2_ and layered double hydroxides (CH@LDHs) as a new sulfur carrier for LSBs. The CH@LDH/S composite delivers a satisfactory first discharge capacity (1014 mAh g^−1^, 0.1 C) and still remains at 653 mAh g^−1^ after 100 cycles. In addition, at 0.5 C, the first discharge capacity is 747 mAh g^−1^, and this specific capacity slowly decays to 491 mAh g^−1^ after 100 cycles. The above results demonstrate that the CH@LDH matrix successfully inhibits the diffusion of LiPSs.

Inspired by the above works, we combined MOF with LDH to design a MOF-derived LDH material with regular three-dimensional (3D) spatial configuration to be used as an ideal sulfur host for high-performance LSBs. Firstly, MIL-88A is used as a template [20], and Ce and Ni ions are added into the alkaline solution for etching. Ce ion doping makes the coordination environment of the MIL-88A polyhedron more stable, especially in the weakly alkaline solution. While the reaction is in progress, urea produces more OH^−^, damaging the MIL-88A polyhedron. With the sustainable precipitation of Ni, Fe, and Ce ions, interconnected nanosheets are formed outside the MIL-88A precursor. Finally, a hollow Ce-Ni/Fe LDH capsule is obtained. The unique structure of Ce-Ni/Fe LDH not only provides the space for the volumetric expansion of the cathode material upon cycling but also supplies abundant polar metal sites to anchor LiPSs and promote the transformation of LiPSs. The electrochemical testing results demonstrate that a lithium–sulfur battery with S@Ce-Ni/Fe LDH delivers the first discharge capacities of 1207 mAh g^−1^ at 0.1 C and 1056 mAh g^−1^ at 0.2 C, respectively. After a long cycle, a satisfactory discharge capacity and an average capacity decay rate per cycle are also achieved (940 mAh g^−1^, 0.067% for 1000 cycles) at 1 C (based on the second cycle).

## 2. Materials and Methods

### 2.1. Synthesis of MIL-88A Polyhedron

Ferric chloride hexahydrate (1.35 g) and fumaric acid (0.58 g) were dissolved in 50 mL deionized (DI) water, respectively. The solution of ferric chloride hexahydrate was quickly added to the solution containing fumaric acid and then poured into a round-bottomed flask and placed in an oil bath at 100 °C for 4 h. At last, the red-brown product was collected and washed with DI water and ethanol, respectively.

### 2.2. Synthesis of Ni/Fe LDH Polyhedron

The obtained precursor (27.5 mg) was ultrasonically dispersed in ethanol (4 mL), and Ni(NO_3_)_2_·6H_2_O (150 mg) and urea (100 mg) were dissolved in DI water (6 mL). Then, two solutions were evenly mixed and sealed in a round-bottomed flask, bathed in an oil bath, and continuously heated at 90 °C for 5 h. During this process, the OH^−^ ions produced by urea hydrolysis gradually corroded the MIL-88A template. The released Fe^3+^, Ni^2+^, and OH^−^ ions were coprecipitated to form a thin shell. The hollow Ni/Fe LDH polyhedron product was rinsed with ethanol and DI water 3 times, respectively, and dried for 12 h at 60 °C.

### 2.3. Synthesis of Ce-Ni/Fe LDH Capsule

The prepared MIL-88A (27.5 mg) was ultrasonically dispersed in ethanol (4 mL). Ni(NO_3_)_2_·6H_2_O (142.5 mg), Ce(NO_3_)_3_·6H_2_O (7.5 mg), and urea (100 mg) were dissolved in DI water (6 mL). Two solutions were evenly mixed and stirred to obtain a new solution. The new solution was placed in a Teflon-lined reactor and continuously reacted at 120 °C for 5 h. The resulting product was rinsed with ethanol and DI water, respectively, and dried overnight at 60 °C to obtain a 3D hollow Ce-Ni/Fe LDH capsule.

### 2.4. Synthesis of S@Ce-Ni/Fe LDH

Ce-Ni/Fe LDH and sublimated sulfur were mixed and ground thoroughly in accordance with the mass ratio of 1:3. The mixture was heated for 12 h at 155 °C under Ar atmosphere. After being cooled to room temperature, the S@Ce-Ni/Fe LDH composite was collected. For comparison, the S@Ni/Fe LDH composite was also prepared by the same method, as shown in Figure 1.

### 2.5. Preparation of Li_2_S_6_ Solution

To test the adsorption capacity of the materials for the LPSs, the Li_2_S_6_ (5 mM) solution was synthesized. S and Li_2_S (5:1, molar ratio) were added to a mixture of 1,3-dioxolane (DOL) and 1,2-dimethoxymethane (DME). At Ar atmosphere, the solution was stirred for 12 h at 60 °C to obtain the Li_2_S_6_ solution.

### 2.6. Materials Characterization

The crystal structures were analyzed by a D/Max-2500/PC X-ray diffractometer (XRD). The morphologies were characterized by JSM-6700F scanning electron microscopy (SEM), field emission SEM (FESEM), scanning transmission electron microscopy (STEM), JEM-2100F transmission electron microscopy (TEM), and high-resolution TEM (HRTEM). The valence states and elements were analyzed by a ESCALAB-250Xi X-ray photoelectron spectrometer. The thermogravimetric test (TGA) was carried out with a NETZSCH STA 499 F3 device. UV–Vis absorption spectra of the adsorbed Li_2_S_6_ solutions were measured by an Evolution 300 UV–Vis spectrophotometer. The N_2_ adsorption/desorption test by a Micromeritics ASAP 2020 analyzer was performed to measure the pore size distribution and specific surface area at 77 K. The pore size distribution and specific surface area were measured by the Barrett–Joyner–Halenda (BJH) method. The wetting angles were determined by the OCA25 contact angle measuring instrument. The molar ratio of the Ni, Fe, and Ce ions in the Ce-Ni/Fe LDH composite was analyzed with an inductive coupled plasma spectrometer.

### 2.7. Electrochemical Measurements

S@Ce-Ni/Fe LDH, acetylene black, and polyvinylidene fluoride were fully ground according to the mass ratio of 70:20:10 and mixed with a N-methyl pyrrolidinone solvent to prepare an electrode slurry, which was evenly coated on Al foil and then dried for 12 h at 60 °C in a vacuum oven to construct the electrode sheet. For comparison, S@Ni/Fe LDH and pure sulfur electrodes were also prepared using the same method. The active sulfur loading in the cathode was approximately 1.0 mg cm^−2^. CR2025 button-type LSBs were assembled in a glovebox filled with an argon atmosphere using different electrodes (S@Ce-Ni/Fe LDH, S@Ni/Fe LDH, and pure S) as the cathode; lithium as the anode; and polypropylene (PP) as the separator. LiTFSI (1 M) in a mixture solvent of DOL and DME (1:1, volume) with 2 wt% LiNO_3_ was utilized as the electrolyte. The cycle performance and rate performance of the assembled LSBs were tested using a charge/discharge tester (LAND CT2001A) with a voltage interval of 1.7–2.8 V (1 C = 1675 mA g^−1^). The reversible capacities of the batteries were calculated according to the weight of active sulfur. Cyclic voltammetry (CV) and electrochemical impedance spectroscopy (EIS) tests were performed at an electrochemical workstation (CHI650D). In the CV, the scan rate was 0.1 mV s^−1^. The voltage in the EIS was 5 mV, and the frequency range was 0.01–10^6^ Hz.

## 3. Results and Discussion

Figure 2a and Appendix A show FESEM images of the MIL-88A precursor. The length and width of the as-synthesized MIL-88A precursor are approximately 4~5 and 1 μm, respectively. It is obvious that the morphology and particle size of the as-synthesized MIL-88A precursor are uniform. Figure 2b and Appendix A display FESEM images of Ce-Ni/Fe LDH. It is seen that the MIL-88A precursor transforms from a polyhedron into a capsule structure with 3D alternately connected layers during the reaction process. Ce ion has excellent polyvalency, agile coordination, and a great affinity for oxygen-containing donors. The growth process can be effectively regulated by introducing a Ce ion-doping form rather than an interface structure form. The crystallographic stability, morphology, and performance of the as-obtained crystals can be optimized [21]. Figure 2c shows the FESEM image of S@Ce-Ni/Fe LDH, which maintains the same morphology and structure as Ce-Ni/Fe LDH. There is sulfur filling in the lamellar shell gap, but more sulfur is integrated into its capsule interior. Thus, this structure has enough space to hold sulfur and effectively alleviates the volumetric change during cycling. At the same time, this structure possesses a large amount of exposed active sites to anchor LiPSs, thus limiting their dissolution and diffusion [22,23]; that is to say, a large number of exposed active surfaces and metal active sites can absorb LiPSs effectively, which provides a strong obstacle to the dissolution and diffusion of LiPSs during charging/discharging processes [24,25]. Figure 2d,e and Appendix A display TEM images of Ce-Ni/Fe LDH. It can be seen that the capsule-like Ce-Ni/Fe LDH shell is composed of lamellated Ce-Ni/Fe LDH, and the inside is a hollow structure. Figure 2f shows a HRTEM image in which a lattice fringe of 0.262 nm corresponds to the (012) crystal plane of Ce-Ni/Fe LDH. In Figure 2g–k, the presence of the Ce, Ni, Fe, and O elements and their spatially homogeneous distribution in the Ce-Ni/Fe LDH capsule are clearly observed. Comparing SEM image of Ni/Fe LDH with the corresponding element mapping results (Appendix A), it is observed that Ce ions have been successfully doped in Ce-Ni/Fe LDH and uniformly distributed in the Ce-Ni/Fe LDH capsule. In Appendix A, it is obvious that the S, Ce, Fe, Ni, and O elements are evenly distributed in S@Ce-Ni/Fe LDH.

Figure 3a shows the XRD patterns of Ni/Fe LDH, Ce-Ni/Fe LDH, and S@Ce-Ni/Fe LDH, and the XRD pattern of the MIL-88A precursor is displayed in Appendix A. The characteristic peaks of the as-synthesized Ni/Fe LDH and Ce-Ni/Fe LDH are consistent with the standard PDF #26-1286. At the same time, the characteristic peaks at 11.8°, 23.1°, 34.6°, and 60.3° correspond to the (003), (006), (012), and (110) planes of Ni/Fe LDH, respectively, proving the successful preparation of Ni/Fe LDH and Ce-Ni/Fe LDH. S@Ce-Ni/Fe LDH, obtained by melting and diffusion reactions with elemental sulfur, shows spiculate diffraction peaks of S_8_, proving S_8_ is in a good crystalline form in the material. The chemical composition and valence of Ce-Ni/Fe LDH were further studied by XPS, and the linear fitting was analyzed (Figure 3b–f). The spectrum in Figure 3b indicates the presence of the Ce, Ni, Fe, and O elements. As shown in Figure 3c, ten peaks are fitted for the Ce 3d high-resolution spectrum. Two binding energy peaks at 872.8 and 878.5 eV are attributed to the 3d_5/2_ orbit of Ce^3+^, and three binding energy peaks at 874.7, 880.2, and 882.7 eV are ascribed to the 3d_5/2_ orbit of Ce^4+^. In addition, two peaks located at 893.6 and 900.4 eV are assigned to the 3d_3/2_ orbit of Ce^3+^, and three binding energy peaks at 897, 903.9, and 909.8 eV are due to the 3d_3/2_ orbit of Ce^4+^ [26,27]. The Fe 2p high-resolution spectrum is presented in Figure 3d. The peaks at 707.3 eV and 721.5 eV are ascribed to the 2p_3/2_ orbit and 2p_1/2_ orbit of Fe^2+^, and the peaks at 712.8 eV and 724.9 eV are assigned to the 2p_3/2_ orbit and 2p_1/2_ orbit of Fe^3+^ [28]. The peak at 718.5 eV is attributed to a satellite peak. It is seen from the above valence analysis of the Fe 2p high-resolution spectrum that Fe in the composite exists in the form of Fe^3+^. Figure 3e shows the Ni 2p high-resolution spectrum. The two peaks at 855.5 and 873.1 eV are Ni 2p_3/2_ and Ni 2p_1/2_, respectively [29], which mainly exist in the form of Ni^2+^. The two peaks at 861.9 and 879.7 eV are attributed to satellite peaks. Based on the valence analysis of the Fe 2p and Ni 2p high-resolution spectra, they are consistent with the existent forms of Ni and Fe in layered bimetallic hydroxide. Figure 3f displays the high-resolution spectrum of O 1s. The three binding energy peaks at 532.9, 531.0, and 529.7 eV belong to a metal–oxygen bond, hydroxide, and oxygen atoms attached to the material surface, respectively [30]. Most of them exist in the form of hydroxide. Combined with the analysis of the Ce 3d, Ni 2p, Fe 2p, and O 1s high-resolution spectra, the successful preparation of Ce-Ni/Fe LDH has been further demonstrated. The molar ratio of the Ni, Fe, and Ce ions in the Ce-Ni/Fe LDH composite is 3.5:1:0.15.

To test the adsorption capacity of Ce-Ni/Fe LDH for LiPSs, we add Ce-Ni/Fe LDH and Ni/Fe LDH to the as-prepared Li_2_S_6_ solutions. As shown in the digital photos in Figure 4a, after 3 h, we observe that the color of the Ni/Fe LDH-Li_2_S_6_ solution changes from yellow to light yellow, while the Ce-Ni/Fe LDH-Li_2_S_6_ solution becomes almost clear and transparent. Therefore, Ce-Ni/Fe LDH has a better adsorption capacity for Li_2_S_6_ than Ni/Fe LDH. Additionally, the UV–Vis spectra are measured before and after the addition of Ce-Ni/Fe LDH and Ni/Fe LDH. Compared with the characteristic peak of the pure Li_2_S_6_ solution, the intensity of the absorption peak of the Ce-Ni/Fe LDH-Li_2_S_6_ solution is reduced significantly, and the abundant metal sites and groups in Ce-Ni/Fe LDH absorb LiPSs through the chemical action, indicating an inhibition effect on the dissolution and diffusion of LiPSs. Appendix A shows the Ce 3d high-resolution XPS spectra of Ce-NiFe LDH and Ce-NiFe LDH-Li_2_S_6_. After interacting with Li_2_S_6_, the peaks of Ce^3+^ display low binding energies, showing the intense chemical interaction between Ce^3+^ and Li_2_S_6_. In Appendix A, the wettability of pure S, S@Ni/Fe LDH, and S@Ce-Ni/Fe LDH electrode sheets to the electrolyte of a lithium–sulfur battery is compared. The wetting angles of pristine S, S@Ni/Fe LDH, and S@Ce-Ni/Fe LDH are 32.2°, 8.5°, and 6.4°, respectively. It is obvious that S@Ce-Ni/Fe LDH possesses the best wetting property and the strongest affinity for the electrolyte. The pore size distribution and specific surface area of Ce-Ni/Fe LDH are studied by the N_2_ adsorption/desorption isothermal curve, and the results are shown in Figure 4b. The specific surface area of Ce-Ni/Fe LDH is 26.192 m^2^ g^−1^, and mesopores account for the vast majority. The effective mesoporous materials expose more active surfaces and afford more sites for the adsorption of LiPSs [31,32,33]. The pore structure existing on the surface of Ce-Ni/Fe LDH can make the sulfur element enter better the laminate shell gap and the inside of the material. Meanwhile, the electrolyte will be easier to infiltrate and contact better with the active material, providing an effective path for ion transmission during cycling, so that the active material in the positive electrode material can be better utilized. The sulfur loading in S@Ce-Ni/Fe LDH is estimated by TGA under a nitrogen atmosphere. In Figure 4c, the mass loss is about 70 wt% below 350 °C. Ce-Ni/Fe LDH remains stable under nitrogen protection; therefore, the mass loading of sulfur is about 70 wt% in S@Ce-Ni/Fe LDH. Figure 4d intuitively displays the inhibition of the shuttling effect by S@Ce-Ni/Fe LDH in a lithium–sulfur battery compared to the shuttling effect in a lithium–sulfur battery with pure sulfur.

To study the electrochemical performance of LSBs, we use different cathodes (S@Ce-Ni/Fe LDH, S@Ni/Fe LDH, and pure S) to assemble and test CR2025 button-type batteries, and the results are presented in Figure 5. Figure 5a displays the initial three CV curves of a lithium–sulfur battery with S@Ce-Ni/Fe LDH at 0.1 mV s^−1^. There are two significant reduction peaks at 2.31 and 2.03 V, ascribed to the solid–liquid–solid transformation of S_8_ → Li_2_S_n_ (4 < n < 8) and Li_2_S_n_ (4 < n < 8) → Li_2_S_2_/Li_2_S, respectively. In addition, the oxidation peaks at 2.34 and 2.44 V are ascribed to the oxidation of Li_2_S/Li_2_S_2_ with a short chain to Li_2_S_n_ (4 < n < 8) with a long chain and the eventual oxidation to S_8_, respectively [34,35,36]. Appendix A indicates the above redox reaction mechanism of a lithium–sulfur battery with S@Ce-Ni/Fe LDH. It is found that the first three CV curves are highly coincident, ensuring that the redox reaction of a lithium–sulfur battery has good reversibility during charging/discharging processes [37,38,39]. Figure 5b shows the charging/discharging curves (1.7–2.8 V) of a lithium–sulfur battery using S@Ce-Ni/Fe LDH as the positive electrode material at different current densities. Fortunately, even with a current density up to 2 C, a lithium–sulfur battery with S@Ce-Ni/Fe LDH still maintains its well-defined plateaus [3,40,41,42], which is in keeping with the CV results (Figure 5a), ensuring the stable operation of the battery under a high current density. Appendix A represents the charging/discharging curves of LSBs with S@Ni/Fe LDH and pure S at different current densities. It is obvious that the reversible capacities of LSBs with S@Ni/Fe LDH and pure S are lower than those of a lithium–sulfur battery with S@Ce-Ni/Fe LDH at the same current densities. Especially, as for a lithium–sulfur battery with pure S, there are no discharge plateaus at 0.3, 0.5, and 1 C, respectively. Appendix A displays the initial charge/discharge curves of LSBs with S@Ce-Ni/Fe LDH, S@Ni/Fe LDH, and pure S at a current density of 0.1 C. The voltage gap (ΔE) between the charge platform and discharge plateau is derived from voltage hysteresis. Compared with LSBs with S@Ni/Fe LDH and pure sulfur, the ΔE value of the lithium–sulfur battery with S@Ce-Ni/Fe LDH is the smallest, indicating the improved redox kinetics. The bar chart in Appendix A illustrates this gap more intuitively. As displayed in Figure 5c, we test the cycle performances of LSBs with S@Ce-Ni/Fe LDH, S@Ni/Fe LDH, and pure S at 0.2 C. The lithium–sulfur battery using S@Ce-Ni/Fe LDH as the cathode material has a first discharge capacity of 1056 mAh g^−1^. The discharge capacity is 759 mAh g^−1^ through 100 cycles, which is better than those of S@Ni/Fe LDH (419 mAh g^−1^) and pure S (300 mAh g^−1^), and the charge/discharge plateaus are still intact (Appendix A). The above results are comparable to the previously reported literatures (Appendix A). Based on the calculations at the beginning of the second cycle, the capacity retention rate reaches a respectable 95%, strongly proving that S@Ce-Ni/Fe LDH inhibits the LiPSs produced during the cycling and increases the utilization rate of the active material. Appendix A demonstrates two peaks in the negative part and two peaks in the positive part, which are consistent with the CV results in Figure 5a.

At the scan rate of 0.1 mV s^−1^, Figure 5d compares the CV results of the LSBs with S@Ce-Ni/Fe LDH, S@Ni/Fe LDH, and pure S. It is observed that the lithium–sulfur battery with S@Ce-Ni/Fe LDH has the sharpest reduction/oxidation peaks, and the distance between the reduction peak and oxidation peak is the smallest; that is, the polarization voltage is the lowest, indicating that it has the best redox kinetics [43,44,45]. As presented in Figure 5e, the rate performances of the LSBs with S@Ce-Ni/Fe LDH, S@Ni/Fe LDH, and pure S are tested. During the 0.1–2 C switching process, the lithium–sulfur battery with S@Ce-Ni/Fe LDH possesses the highest specific capacity, and the initial discharge capacity reaches 1207 mAh g^−1^ at 0.1 C. After the conversion process of different current densities and 60 cycles (returning to 0.1 C), it still maintains the discharge capacity of 690 mAh g^−1^. Even at the high current density of 2 C, the lithium–sulfur battery with S@Ce-Ni/Fe LDH has still a discharge capacity of 489 mAh g^−1^. However, the discharge capacities in the LSBs with pure S and S@Ni/Fe LDH decrease rapidly when switching to the higher current density, which demonstrates that the polarization is serious under the high current density, fully indicating that the lithium–sulfur battery with S@Ce-Ni/Fe LDH possesses the best rate performance. The EIS spectra of the LSBs with S@Ce-Ni/Fe LDH, S@Ni/Fe LDH, and pure S are analyzed in Figure 5f. Each curve is made up of a semicircle and a slanted straight line, which are attributed to the charge transfer resistance (R_ct_) derived from the diffusion of Li^+^ and Warburg impedance (Z_w_), respectively [46,47], as shown in an equivalent circuit diagram in Appendix A. In the equivalent circuit diagram, R_e_ is the resistance of the electrolyte, and CPE stands for the interfacial capacitance [48]. The fitting of the EIS spectra of the LSBs with S@Ce-Ni/Fe LDH, S@Ni/Fe LDH, and pure S before cycling is shown in Appendix A. According to the fitting results, among the three LSBs, the lithium–sulfur battery with S@Ce-Ni/Fe LDH has the lowest R_ct_ value (56 Ω), which is significantly lower than those of S@Ni/Fe LDH (75 Ω) and pure S (90 Ω). In addition, compared with the LSBs with S@Ni/Fe LDH and pure S, the lithium–sulfur battery with S@Ce-Ni/Fe LDH has the highest slope in the low-frequency region, demonstrating its fastest diffusion rate of Li^+^ [49,50]. As shown in Figure 5g, the lithium–sulfur battery with S@Ce-Ni/Fe LDH is tested at 1 C for 1000 cycles. The battery is first activated at 0.1 C for five cycles, and then, the current density is returned to 1 C. The first specific capacity still remains at 940 mAh g^−1^. After 1000 cycles, the specific capacity is kept at 308 mAh g^−1^, and the average capacity attenuation rate per cycle is 0.067% (based on the second cycling). Appendix A presents a digital photo of a light-emitting diode (LED) green lamp string lit up by two CR2025-type LSBs with S@Ce-Ni/Fe LDH connected in a series, fully showing the practical application prospect of this work for high-performance LSBs.

## 4. Conclusions

We successfully designed and prepared a capsule-type S@Ce-Ni/Fe LDH as a cathode material for LSBs, in which the 3D hollow Ce-Ni/Fe LDH is derived from the MIL-88A polyhedron. The unique capsule-like structure of Ce-Ni/Fe LDH possesses enough space to hold sulfur and a large amount of exposed active sites, which play an important role in preventing the dissolution and diffusion of LiPSs. Compared to the lithium–sulfur battery with the traditional sulfur cathode, the electrochemical properties of the lithium–sulfur battery with S@Ce-Ni/Fe LDH were significantly improved. The first discharge capacities of the lithium–sulfur battery with S@Ce-Ni/Fe LDH were 1207 and 1056 mAh g^−1^ at 0.1 and 0.2 C, respectively. Even at a large current density of 1 C, the lithium–sulfur battery with S@Ce-Ni/Fe LDH also delivered an initial discharge capacity of 940 mAh g^−1^ and could cycle 1000 times, with an average capacity decay rate per cycle of 0.067% (based on the second cycle). This work provided a new and simple idea to design a novel cathode carrier for high-performance LSBs. At the same time, the obtained active material supplies a direction for the development of other energy storage fields, such as room temperature Na-S batteries.

## Figures and Tables

**Figure 1 nanomaterials-13-02244-f001:**
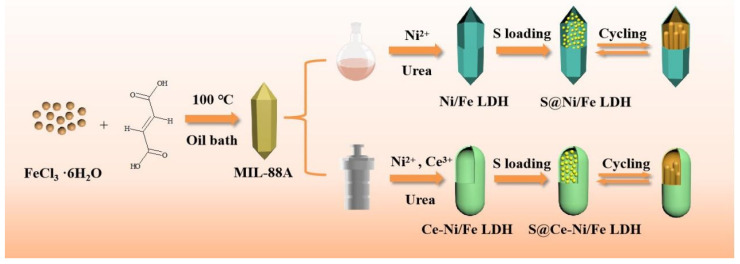
Schematic diagram of the synthesis of S@Ni/Fe LDH and S@Ce-Ni/Fe LDH.

**Figure 2 nanomaterials-13-02244-f002:**
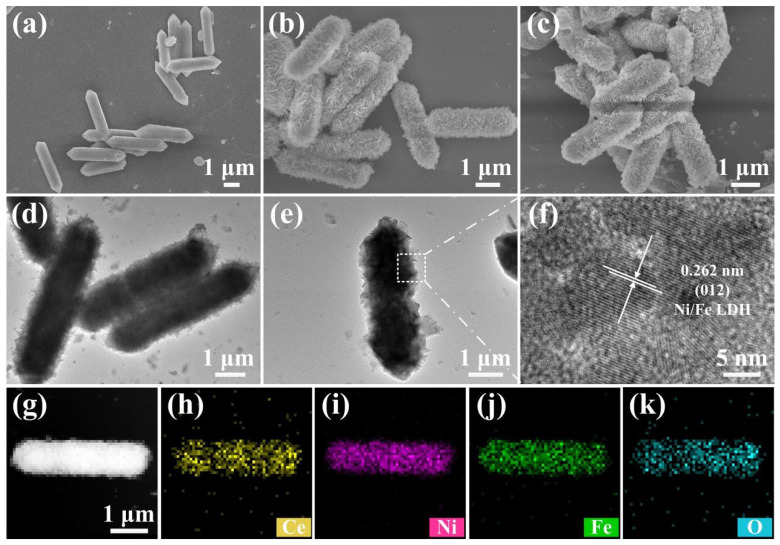
FESEM images of (**a**) the MIL-88A precursor, (**b**) Ce-Ni/Fe LDH, and (**c**) S@Ce-Ni/Fe LDH. (**d**,**e**) TEM and (**f**) HRTEM images of Ce-Ni/Fe LDH. (**g**) STEM image of Ce-Ni/Fe LDH, and (**h**–**k**) the corresponding elemental mappings of Ce, Ni, Fe, and O.

**Figure 3 nanomaterials-13-02244-f003:**
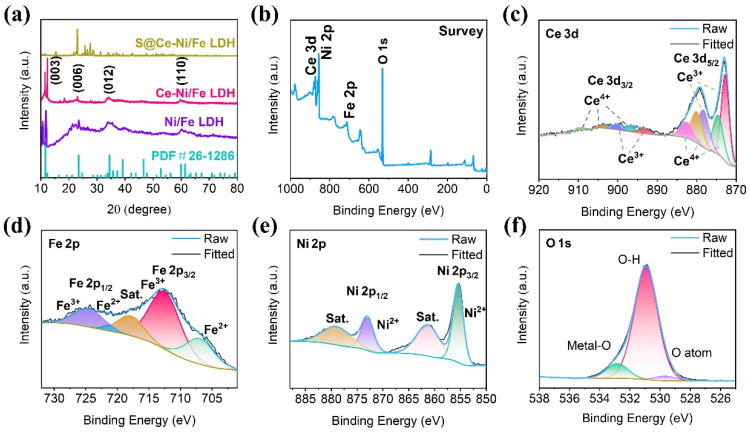
(**a**) XRD patterns of Ni/Fe LDH, Ce-Ni/Fe LDH, and S@Ce-Ni/Fe LDH. (**b**) XPS survey spectrum of Ce-Ni/Fe LDH. High-resolution XPS spectra of (**c**) Ce 3d, (**d**) Fe 2p, (**e**) Ni 2p, and (**f**) O 1s of Ce-Ni/Fe LDH.

**Figure 4 nanomaterials-13-02244-f004:**
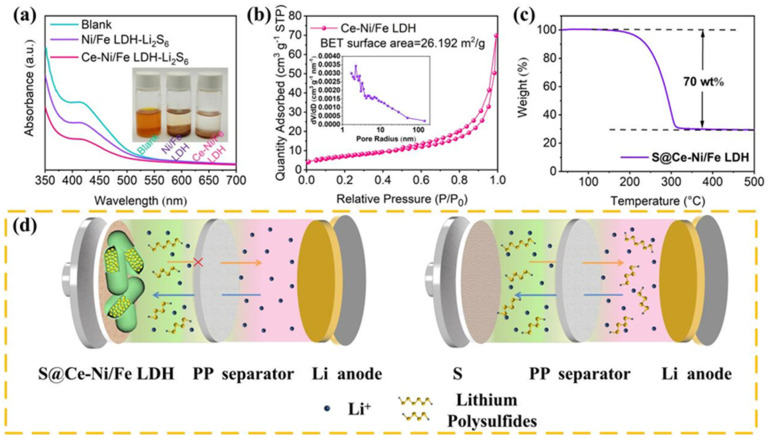
(**a**) UV–Vis spectra and digital photos of pure Li_2_S_6_ and Li_2_S_6_ solutions after adding Ni/Fe LDH and Ce-Ni/Fe LDH. (**b**) N_2_ adsorption/desorption isotherm, and the pore size analysis (inset) of Ce-Ni/Fe LDH. (**c**) TGA curve of S@Ce-Ni/Fe LDH. (**d**) Schematic diagram of the suppression of the shuttling effect in a lithium–sulfur battery with S@Ce-Ni/Fe LDH, and a schematic diagram of the shuttling effect in a lithium–sulfur battery with pure sulfur.

**Figure 5 nanomaterials-13-02244-f005:**
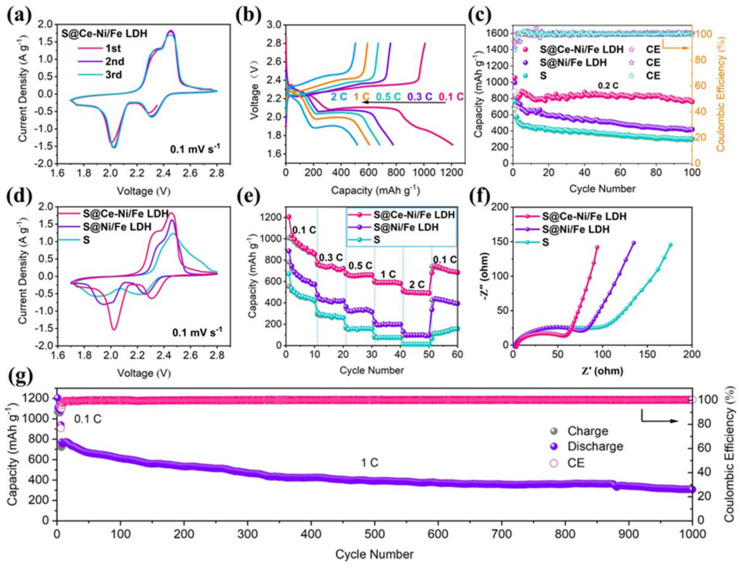
(**a**) CV curves of the lithium–sulfur battery with S@Ce-Ni/Fe LDH during the initial three cycles at 0.1 mV s^−1^. (**b**) Charge/discharge curves of the lithium–sulfur battery with S@Ce-Ni/Fe LDH at different current densities. (**c**) Cycling performance and CE of the LSBs with S@Ce-Ni/Fe LDH, S@Ni/Fe LDH, and pure S for 100 cycles at 0.2 C. (**d**) CV curves at 0.1 mV s^−1^, and (**e**) the rate performances of the LSBs with S@Ce-Ni/Fe LDH, S@Ni/Fe LDH, and pure S. (**f**) EIS spectra of the LSBs with S@Ce-Ni/Fe LDH, S@Ni/Fe LDH, and pure S before cycling. (**g**) Long-term cycling stability and CE of the lithium–sulfur battery with S@Ce-Ni/Fe LDH for 1000 cycles at 1 C.

## Data Availability

Not available.

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
