# Peer review of "Ce-Doped Three-Dimensional Ni/Fe LDH Composite as a Sulfur Host for Lithium–Sulfur Batteries"

_nanomaterials, 2023, doi:10.3390/nano13152244_

Round 1
Reviewer 1 Report
This manuscript presents properties of Ce-doped three-diemensional Ni/Fe LDH composites as a sulfur host for lithium sulfur batteries. The typical problem of lithium sulfur batteries that is shuttling of lithium polysulfide during its operation is solved by the rationally designed material. Especially, both hollow structure and Ce-doped Ni/Fe LDH effectively traps lithium polysulfides and thus inhibit the shuttling phenomenon. All of the experimental results of this manuscript well described and well-oriented. Therefore, this manuscript is acceptable in this journal after answering the following comments and questions.
(1) The relative molar ratio between metal species including Ni, Fe, and Ce should be analyzed by another analytical method such as ICP.
(2) What is peak observed close to 10o from Ni/Fe LDH and Ce-Ni/Fe LDH? It seems the peak is not related to the reference LDH structure. The peak position of the un-defined peak from Ni/Fe LDH and Ce-Ni/Fe LDH seems not identical.
(3) Like XPS signal analysis done for Fe 2p, labeling of each signal from Ce3d, Ni2p, O1s should be indicated in each figure.
(4) The charge/discharge curves of lithium-sulfur battery with different current densities of S, S@Ni/Fe LDH should be provided in Supplementary files.
(5) To clarify the possible reaction mechanisms of the lithium-sulfur battery in different samples studied in this manuscript, the reaction mechanism mentioned in the text can be added as insets in Figure 5a.
(6) To make sure that the S@Ni/Fe LDH has better stability toward shuttling of polysulfide, the charge and discharge profiles and corresponding dQ/Dv Plots at higher cycle number (i.e @100th cycle) should be plotted and displayed.
(7) Even though S@Ce-Ni/Fe LDH has higher capacity and improved polarization, it seems the material also has drastic capacity fading when it is cycled at 0.1C up to 10th cycle as shown in Figure 5e. What is the main reasons for the capacity fading?
(8) Authors mentioned that the LDH has proved to be stabilized toward polysulfide shuttling phenomenon in the introduction part. The electrochemical properties such as reversible capacity, rate capability, and/or cycling stability of S@Ce-Ni/Fe LDH should be compared with previously reported results.
(9) It seems the role and effect of Ce in the LDH for stabilizing polysulfide shuttling are not clear. Can authors provide more evidence that the Ce has clear effect toward stabilizing the polysulfide shuttling?
There are no clear problems in English for this manuscript.
Reviewer 2 Report
As a cathode material for LSBs, the authors have successfully created a capsule-type S@Ce-Ni/Fe LDH. This material has a large number of exposed active sites and enough space to hold sulphur, both of which are crucial in preventing the dissolution and diffusion of LiPSs. The electrochemical performance of a lithium-sulfur battery with S@Ce-Ni/Fe LDH has greatly improved compared to a lithium-sulfur battery with a conventional sulphur cathode. However, the author must address a few minor concerns before publishing.
1. In light of new studies on the LSBs, the author needs to strengthen the introduction.
2. What percentage of S is loaded in this study? Additionally, EDS of S@Ce-Ni/Fe LDH should be performed to check for S homogeneity in the structure.
3. Why S@Ce-Ni/Fe LDH is shown to be activated during the first few cycles in Figure 5(c)? Additionally, Figure 5(e) should explain why S@Ce-Ni/Fe LDH retention is lower.
4. A comparison of the most recent study with a few references would be beneficial.
Reviewer 3 Report
I have read this interesting work on the promising new design approach of Lithium Sulfur batteries, with a metal organic framework-derived and Ce-ion doped LDH (Ce-Ni/Fe LDH) with hollow capsule configuration.
I believe this approach could provide some new insight in new better batteries.
However, prior to publication I would like the authors to comment on two important issues :
1. How big is the ecological impact and carbon imprint of the new batteries compared to the traditional e.g. LiPo or or other comparable ones ?
2. How about recyclability issues ? Will it be easy to produce a fully recyclable batteries series based on their proposed technologies ?
3. Please post a table of main characteristics against other major battery types , of the proposed new battery (comparable categories)
I have read this interesting work on the promising new design approach of Lithium Sulfur batteries, with a metal organic framework-derived and Ce-ion doped LDH (Ce-Ni/Fe LDH) with hollow capsule configuration.
I believe this approach could provide some new insight in new better batteries.
However, prior to publication I would like the authors to comment on two important issues :
1. How big is the ecological impact and carbon imprint of the new batteries compared to the traditional LiPo ones ?
2. How about recyclability issues ? Will it be easy to produce a fully recyclable batteries series based on their proposed technologies ?
3. Please post a table of main characteristics against other major battery types , of the proposed new battery (comparable categories)
Reviewer 4 Report
Ce-doped three-dimensional Ni/Fe LDH composite as a sulfur host for lithium-sulfur batteries
1. The synthesis process is confusing. In some works, it is found that some mof can easily form LDHs with metal solution under hydrothermal treatment. Why did authors use urea? A compete mechanism for the synthetic process should be explained in detail.
2. The rational use of MIL based MOF can be explained with the following articles:
Separation and Purification Technology, Volume 287, 15 April 2022, 120463
3. Authors need deep literature review for MOFs and LDHs based materials with the reference of following articles:
Journal of Energy Storage, Volume 60, April 2023, 106713, Journal of Energy Storage, Volume 72, Part A, 15 November 2023, 108220
4. EIS fitting is necessary.
Minor editing of English language required
